# IL-6 Serum Levels Can Enhance the Diagnostic Power of Standard Blood Tests for Acute Appendicitis

**DOI:** 10.3390/children9101425

**Published:** 2022-09-20

**Authors:** Marco Di Mitri, Giovanni Parente, Giulia Bonfiglioli, Eduje Thomas, Cristian Bisanti, Chiara Cordola, Marzia Vastano, Sara Cravano, Edoardo Collautti, Annalisa Di Carmine, Simone D’Antonio, Tommaso Gargano, Michele Libri, Mario Lima

**Affiliations:** Paediatric Surgery Department, IRCCS Sant’Orsola-Malpighi University Hospital, 40138 Bologna, Italy

**Keywords:** acute appendicitis, complicated acute appendicitis, interleukine-6, peritonitis

## Abstract

Background: The diagnosis of acute appendicitis (AA) remains challenging, especially in pediatrics, because early symptoms are not specific, and the younger the patient the more difficult their interpretation is. There is a large degree of agreement between pediatric surgeons on the importance of an early diagnosis to avoid complicated acute appendicitis (CAA) and its consequences. The aim of this study is to assess if Interleukin 6 (IL-6) could enhance the sensitivity (Sn) and specificity (Sp) of the currently available and routinely performed diagnostic tools in case of suspected AA in pediatric patients. Materials and Methods: A prospective observational study was conducted including patients who underwent appendectomy between November 2020 and March 2022. We divided patients into three groups: not inflamed appendix (group NA), not complicated AA (group NCAA), and complicated AA (group CAA). We compared the mean values of white blood cells (WBC), neutrophils, fibrinogen, ferritin, aPTT, INR, C-reactive protein (CRP), IL-6, and CRP between the three groups. Then we evaluated Sn, Sp, and odds ratio (OR) of IL-6 and CRP alone and combined. Results: We enrolled 107 patients operated on for AA (22 in Group NA, 63 in Group NCAA, and 21 in group CAA). CRP levels resulted in a significant increase when comparing CAA with NA (*p* = 0.01) and CAA with NCAA (*p* = 0.01), whereas no significance was found between NA and NCAA (*p* = 0.38). A statistically significant increase in average IL-6 levels was found when comparing NCAA with NA (*p* = 0.04), CAA with NA (*p* = 0.04), and CAA with NCAA (*p* = 0.02). Considering CRP alone, its Sn, Sp, and OR in distinguishing NA from AA (both NCAA and CAA together) are 86%, 35%, and 33,17, respectively. Similarly, Sn, Sp, and OR of IL-6 alone are 82%, 54%, and 56, respectively. Combining CRP and IL-6 serum levels together, the Sn increases drastically to 100% with an Sp of 40% and OR of 77. Conclusions: Our study may suggest an important role of IL-6 in the detection of AA in its early stage, especially when coupled with CRP.

## 1. Introduction

Acute appendicitis (AA) is the most common cause of abdominal pain in children, with incidence varying from 1 to 6/10,000 between 0- and 4-year-old children and from 19 to 28/10,000 between 5 and 14 years of age. Boys are more frequently affected than girls [1,2].

AA can quickly evolve to peritonitis in children; therefore, early diagnosis is fundamental.

The most frequent cause of AA is appendiceal lumen obstruction by appendicoliths followed by lymphoid hyperplasia, parasites, foreign bodies, and tumors [3,4,5]. 

The obstruction of the lumen leads to the distension of the appendix, the latter worsened by physiological mucus production, followed by bacterial overgrowth, vascular impairment, necrosis, and perforation [3,6].

The diagnosis of AA remains challenging, especially in pediatrics, because early symptoms are not specific, and the younger the patient the more difficult their interpretation. 

Even if it is quite common, AA is confirmed to be one of the most misdiagnosed surgical emergencies.

Symptoms of AA in children include abdominal pain starting as periumbilical and then moving to the right lower side worsening as time goes on and causing loss of appetite, emesis, and fever.

However, young children may lack this history of progression and migration of pain arriving at the emergency department with symptoms of early perforation or at worst sepsis.

When the patient presents the typical clinical pattern, AA can be easily diagnosed. On the contrary, more investigations are needed such as blood tests, abdomen ultrasounds, and, sometimes, abdomen CT scans [7].

When performing blood tests, white blood cells (WBC), especially neutrophils, and C-reactive protein (CRP) are usually high, with a sensitivity (Sn) of 79%, 95%, and 46%, respectively, in uncomplicated acute appendicitis (NCAA). In the case of complicated acute appendicitis (CAA), the sensitivity of such markers increases to 84%, 84%, and 99%, respectively [8]. Likewise, the reported specificity (Sp) of these markers, routinely screened in patients with the suspicion of AA, is 44–62,5% for WBC and 88–100% for CRP [9].

Laparoscopic appendectomy has become the gold standard treatment, replacing the conventional open technique [10,11,12].

Even if the best treatment and its timing are still debated (medical treatment and delayed appendectomy vs. emergent appendectomy), there is a large degree of agreement between pediatric surgeons on the importance of an early diagnosis to avoid complicated acute appendicitis and its consequences [13].

In literature, the reported negative appendectomy rate varies from 4 to 45%. Many scoring systems, biochemical markers, and radiological imaging have been studied to decrease unnecessary appendectomies that are not free from complications [14,15,16,17,18].

In this prospective study, we analyzed the diagnostic value of the Interleukin 6 (IL-6), a pro-inflammatory cytokine mainly produced by macrophages during acute inflammation, in comparison with standard blood markers dosed for suspected AA. Due to the early increase in IL-6 during the acute phase response due to bacterial endotoxemia, we hypothesized that it may have a role in the early diagnosis of AA, whereas before it had been never taken into account [19].

The aim of this study is to assess if IL-6 could enhance the sensitivity and specificity of the currently available and routinely performed diagnostic tools in case of suspected AA in pediatric patients. 

## 2. Materials and Methods

A prospective observational study was conducted at our Department of Pediatric Surgery, IRCCS Sant’Orsola-Malpighi, *Alma Mater Studiorum*, University of Bologna (IT), University Hospital of Bologna, following Ethical Committee approval with Ethic Code CHPED-22-03-AAC (25 May 2022).

We enrolled 107 patients who underwent appendectomy between November 2020 and March 2022.

Inclusion criteria were: being a pediatric patient (under 16 years of age), presenting signs and symptoms suggestive of acute appendicitis, and having undergone blood exams with IL-6 too. 

Exclusion criteria were: patients not operated on for AA and ones with AA that did not have IL-6 serum dosage performed.

Personal data, such as age at surgery and sex, medical history, signs, and symptoms when patients were referred to the emergency department, blood tests (WBC, neutrophils, INR, aPTT, fibrinogen, ferritin, CRP), and histological reports on the appendices removed were revised. Moreover, the IL-6 serum level was dosed in all patients together with other inflammatory markers at the time when blood tests were performed in the emergency department.

Informed consent was obtained from parents or legal guardians once the patient had been admitted to the emergency department and the decision to investigate for acute appendicitis had been taken.

The primary outcome of the study was to investigate whether IL-6 may have a role in the diagnostic work-up of AA.

The secondary outcome was to verify if the combination of IL-6 with the routinely performed biomarkers may reduce the number of negative appendectomies. 

These inflammatory markers were acute phase reactants and could be elevated in all processes in which a local or systemic inflammation was triggered. In our study, only patients presenting isolated appendicitis were considered. All patients with concomitant inflammatory processes in other districts were excluded.

We divided patients into three groups based on the intraoperative findings and histological reports: not inflamed appendix (group NA 23/107—21.5%), not complicated AA (group NCAA 63/107—58.9%), complicated AA (group CAA 21/107—19.6%) defined as a gangrenous or perforated appendix with/without abscess and peritonitis.

In the study design, healthy controls were not included because our focus was centered on patients with the suspicion of an infective process regarding the appendix. Due to the existence of validated standard values defining the normality range of inflammatory markers, these were taken as a reference to determine the degree of inflammation.

We then compared the mean values of WBC, neutrophils, fibrinogen, ferritin, aPTT, INR, CRP, IL-6, and CRP between the three groups. Then we evaluated Sn, Sp, and odds ratio (OR) of IL-6 and CRP alone and combined.

Continuous data are reported as mean ± standard deviation. Statistical analysis was performed using Student’s *t*-test after verifying the normal distribution of variables with the Shapiro–Wilk test; a *p*-value < 0.05 was considered statistically significant.

## 3. Results

Between November 2020 and March 2022, 228 patients underwent appendectomies at our Pediatric Surgery Department. A total of 121 patients did not undergo IL-6 dosage because blood tests were already performed in other emergency departments before referring them to us and parents did not give their consent for another blood sample. 

Therefore, 107/228 patients were enrolled in the study: 69 males (64.5%) and 39 females (35.5%). The mean age at surgery was 10.3 ± 2.5 (range: 4–16) years and divided into the following groups:

We divided patients into three groups based on the intraoperative findings and histological reports:-NA: 23/107 (21.5%).-NCAA: 63/107 (58.9%).-CAA: 21/107 (19.6%).

WBC mean values were significantly increased when comparing NCAA with NA (*p* = 0.03) and CAA with NA (*p* = 0.04), whereas no significant differences were found between NCAA and CAA (*p* = 0.4).

Fibrinogen mean values were significantly higher when comparing CAA (*p* < 0.01), NCAA (*p* < 0.01), and CAA with NA (*p* < 0.01); on the contrary, no statistically significant differences were found between NA and NCAA (*p* = 0.4).

Mean values of neutrophils, aPTT, INR, and ferritin did not show any statistically significant differences comparing the three groups (data are summarized in Table 1). 

CRP levels significantly increased when comparing CAA with NA (*p* = 0.01) and CAA with NCAA (*p* = 0.01), whereas no significance was found between NA and NCAA (*p* = 0.38).

A statistically significant increase in average IL-6 levels was found comparing NCAA with NA (*p* = 0.04), CAA with NA (*p* = 0.04), and CAA with NCAA (*p* = 0.02).

Considering CRP alone, its Sn, Sp, and OR in distinguishing NA from AA (both NCAA and CAA together) are 86%, 35%, and 33,17, respectively. 

Similarly, Sn, Sp, and OR of IL-6 alone are 82%, 54%, and 56, respectively.

Combining CRP and IL-6 serum levels together, the Sn increases drastically to 100% with an Sp of 40% and OR of 77 (Figure 1).

## 4. Discussion

The diagnosis of AA is challenging, especially in young children where symptoms are not specific and hardly reported by patients.

A key point of the diagnosis, except for the physical examination, is blood sampling to determine mainly WBC, neutrophils, and CRP serum levels. 

Many studies analyzed these parameters as predictive factors of AA. 

Glass et al. in their literature review about blood tests and AA diagnosis reported the following remarkable results: WBC has an Sn of 70–80% and Sp of 60–68%, the absolute neutrophil count has an Sn of 59–97% and Sp of 51–90%, CRP has an Sn of 58–93% and Sp of 28–82% [7].

Chung et al. showed how in perforated AA CRP main values were statistically increased compared with patients with not complicated AA (92 mg/L vs. 31 mg/L, *p* < 0.01) [14].

Nevertheless, several studies showed how the above-reported parameters have a low predictive value if considered alone [20,21,22]. To overcome the poor predictive value of the parameters taken singularly, a lot of efforts have been made to develop multiparameter diagnostic scores, considering both symptoms and blood tests to estimate the risk of AA in pediatric patients. Samuel et al. validated the PAS—pediatric appendicitis score—that considers parameters such as pain migration, anorexia, vomit, pain to the right iliac fossa, cough, fever, leukocytosis, and neutrophil count. Analyzing 1170 patients, they reported a 100% Sn and 92% Sp for a PAS score ≥ 6 (*p* < 0.01) [23].

On the contrary, Attia et al., applying PAS score in a prospective study on 849 pediatric patients, showed only a 72% Sn and a 94% Sp [24].

Another study validated the Alvarado score, which uses a combination of parameters such as migration pain in the right iliac fossa, anorexia, vomit, pain in the right iliac fossa, Blumberg’s sign, fever, leukocytosis, and neutrophil count. They enrolled 305 patients reporting a main score of 7.7 in patients with confirmed AA and 5.2 in patients without AA. They concluded that AA is unlikely with a score < 3. Patients with a score between 5–6 require observation and re-evaluation whereas patients with a score ≥ 7 should be referred to pediatric surgeons [25]. The Alvarado scores of Sn and Sp were reported as 72% and 81%, respectively, by Schneider et al. in a prospective study on 588 patients [26].

Another score is the AIR—appendicitis inflammatory response—that considers parameters such as vomit, pain in the right iliac fossa, pain degree, fever, neutrophil count, leukocytosis, and CRP. In a multicentric study on 3878 patients with suspected AA, Manne Andersson et al. showed a high probability of AA, with a predictive positive value (PPV) of 96%, when the AIR score was > 8 (*p* < 0.01) [27].

Lima et al., in a retrospective study on 1025 patients, developed the APPE score—APpendicitis PEdiatric score—that includes the evaluation of age, sex, time from onset of symptoms, vomit, right iliac fossa, Blumberg’s sign, WBC, neutrophil percentage, and CRP. The maximum score was 21 points and they showed how patients with a score ≤ 8 have a low risk of AA, and those with a score between 8 and 15 had intermediate risk, whereas those ≥ 15 had a high risk of AA with Sn = 94% and Sp = 93% [28]. 

All the previously cited scores share the ability to diagnose complicated AA but lack the early diagnosis of AA. 

Consequently, research is now focused on pro-inflammatory mediators and cytokines secreted during the early stage of inflammation. 

Some authors suggest IL-6 as a marker to be included in an early diagnostic algorithm of AA in pediatrics. Among these, a prospective study conducted by Kakar et al. divided 92 pediatric patients into three groups: control group, complicated acute appendicitis, and not complicated acute appendicitis. Urine and serum samples were collected to evaluate IL-6 and NGAL (neutrophil gelatinase-associated lipocalin) levels the day before surgery, and on the 2nd and 5th postoperative day, respectively. Kakar et al. demonstrated that only serum IL-6 was statistically higher in both complicated and not complicated AA compared with the control group showing an Sn of 77.4% and an Sp of 58.1% (*p* < 0.01) [29].

Haghy et al., in their prospective study, enrolled patients with suspected AA and an Alvarado score > 5. Procalcitonin (PCT) and IL-6 serum levels were tested, demonstrating a statistically significant reduction in unnecessary appendectomies by adding both markers to the diagnostic work-up. PCT showed an Sn and Sp of 65% and 80%, respectively, suggesting a possible role in AA diagnosis. Finally, the authors highlighted how simultaneous high levels of PCT and IL-6 achieve an Sn of 95% and Sp of 55% [30].

Ozguner et al. demonstrated how the serum IL-6 level was significantly lower (*p* < 0.01) in children with an intraoperative finding of not inflamed appendix compared with patients with complicated/uncomplicated AA concluding that IL-6 dosage may reduce the number of unnecessary appendectomies (Sn: 76%, Sp: 81%) [31]. 

Gurleyik et al. reported an Sn of 84% and Sp of 46% of IL6 in a study in which 77 patients were enrolled, declaring how an important increase in serum IL-6 levels was found in patients with acute perforated appendicitis. Nevertheless, a high number of false positives (54%) and false negatives (16%) associated with the use of serum IL-6 levels was reported. Considering these results, they stated that serum IL-6 levels alone do not bring any improvement in the accuracy of AA diagnosis [32]. 

Similar conclusions were drawn by Goodwin et al., analyzing 53 patients with left lower abdominal pain where a statistically significant difference between the IL-6 level in AA and a not inflamed appendix was not found [33].

In contrast with studies complaining about a lack of accuracy of IL-6 in the diagnosis of AA, our study showed that this interleukin can be a helpful biomarker if associated with CRP. CRP alone, in our study, could distinguish between NCAA and CAA (*p* = 0.01) and between CAA and NA (*p* = 0.01) but lacked the ability to distinguish between NA and NCAA. 

On the contrary, IL-6 showed the same characteristic as CRP but could also distinguish between NA and NCAA (*p* = 0.04), suggesting the ability to detect AA even in the early stage. Furthermore, we analyzed Sn and Sp of IL-6 and CRP reporting 82% and 54% for IL6 and 86% and 35% for CRP, respectively. These results are comparable with ones in the literature.

Finally, we studied the diagnostic power of CRP and IL-6 taken together, showing a surprising Sn of 100% and Sp of 40%.

Recently, some studies have investigated the sensitivity of other laboratory markers in detecting the presence of AA in children. Anand et al. reviewed the literature on red cell distribution width and mean platelet volume in separate systemic reviews, without identifying a clear role in diagnosing pediatric AA [34,35,36], whereas the pooling data showed significantly elevated levels of pentraxin-3 in AA compared to healthy controls [34]. In another meta-analysis, the same authors highlighted hyponatremia as an independent prognosticator of CAA [18].

### Limitations of the Study

We are conscious of the limits of this study. First, the low number of enrolled patients decreases the statistical significance of our conclusions. Nevertheless, we think our data could be an interesting and encouraging preliminary result for further investigations such as multicentric studies.

Moreover, we are aware that enrolling only patients who underwent appendectomy could represent a selection bias. Additionally, the decision not to operate on a patient was taken mostly based on a physical examination not compatible with AA; thus, blood tests wouldn’t have changed the clinicians’ choices.

At last, we must consider the existence of a wide spectrum of conditions in children in which IL-6 might be elevated due to the presence of local or systemic inflammation. This potential bias compelled us to exclude patients with other known concomitant medical conditions to minimize any changes in the inflammatory markers not related to AA. However, we are conscious that such a choice might not be sufficient to completely avoid selection bias, and some inflammatory conditions could be undiagnosed.

## 5. Conclusions

The early diagnosis of AA remains challenging. Several studies analyzed the role of IL-6 in the diagnosis of AA. Our study may suggest the important role of IL-6 in the detection of AA in its early stage, especially when coupled with CRP. This is even more important, considering the number of unnecessary appendectomies that can be avoided when enhancing the diagnostic power of blood tests.

However, we are aware that IL-6 is not specific for AA and we are therefore far from considering it unerring. Nevertheless, we endorse its diffusion, enhancing the diagnostic power of the traditional blood tests and physical examination and/or ultrasound.

## Figures and Tables

**Figure 1 children-09-01425-f001:**
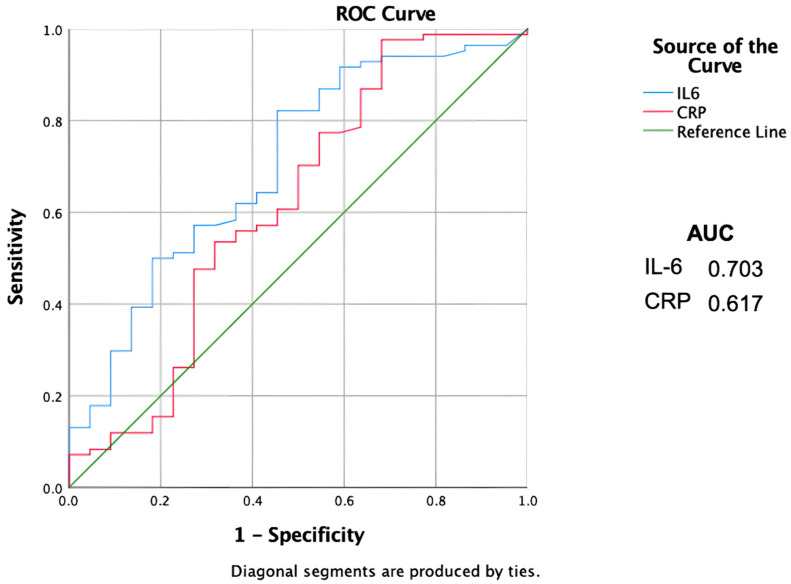
ROC curves showing sensitivity and specificity of the inflammatory biomarkers. AUC = Area Under the Curve. Il-6 = Interleukin 6. CRP = C-reactive protein.

**Table 1 children-09-01425-t001:** Mean values, standard deviation (SD) and *p*-value of white blood cells (WBC), neutrophils (Neu), aPTT, INR, fibrinogen (Fib), ferritin (Fn), C-reactive protein (CRP), and Interleukine-6 (IL-6). NA = Not inflamed appendix. NCAA = not complicated acute appendicitis. CAA = complicated acute appendicitis. * = Statistically significant values.

	n	WBC (10^9/L)	Neu (%)	aPTT (s)	INR	Fib (mg/dL)	Fn (ng/mL)	CRP (mg/dL)	IL-6 (mg/dL)
**NA**	23	11.5 ± 3.9	69.4 ± 16.4	1.09 ± 0.15	1.19 ± 0.08	367 ± 115.3	49 ± 33.6	4 ± 5.8	20 ± 15.1
**NCAA**	63	13.9 ± 5.3	74.5 ± 13.2	1.10 ± 0.17	1.18 ± 0.09	374 ± 86	54 ± 31.2	4.5 ± 7.9	30 ± 65
**CAA**	21	15.5 ± 7.3	78 ± 18.5	1.06 ± 0.15	1.2 ± 0.1	497 ± 82	187 ± 315.8	9 ± 8.1	394 ± 780
** *p-Values* **									
**NA vs. NCAA**		0.03 *	0.2	0.4	0.4	0.4	0.3	0.38	0.04 *
**NA vs. CAA**		0.04 *	0.12	0.27	0.23	0.01 *	0.06	0.01 *	0.04 *
**NCAA vs. CAA**		0.4	0.44	0.16	0.14	0.01 *	0.06	0.01 *	0.02 *

## Data Availability

The data presented in this study are available on request from the corresponding author. The data are not publicly available due to privacy restrictions.

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
