# Peer review of "IL-6 Serum Levels Can Enhance the Diagnostic Power of Standard Blood Tests for Acute Appendicitis"

_children, 2022, doi:10.3390/children9101425_

Round 1

Reviewer 1 Report

At the outset, I would like to congratulate the authors on a well-conducted study. The research subject is interesting and will be of interest to our readers. However, I have several comments that need to be addressed.  

Introduction: Please add your hypothesis in 2-3 lines at the end of the Introduction section. What did you hypothesize before conducting this study?

Methods and Results: It is always better to represent the comparative biomarker accuracy by using ROC curves. Please provide ROC curves for the biomarkers studied. 

-IL6, CRP, and Fibrinogen are non-specific markers and could be elevated in many systemic diseases. Adequate adjustment for confounding was not performed in this study. This could bias the results. Was this taken into account?

-Why were healthy controls not included in the study?

Discussion:

-Please provide a biological explanation for rising levels of these biomarkers in appendicitis

-Few systematic reviews and meta-analyses have been published on recent biomarkers in appendicitis. Please discuss them in the discussion section.

https://link.springer.com/article/10.1007/s00383-022-05149-4

https://pubmed.ncbi.nlm.nih.gov/35884054/

https://pubmed.ncbi.nlm.nih.gov/35885500/

https://pubmed.ncbi.nlm.nih.gov/35454059/

Minor

There are multiple grammatical errors:

-Spelling mistake line 118- "CPR levels" should have been "CRP levels"

-grammar error line 223

Please take the help of a writing assistant.

Reviewer 2 Report

In the introduction, you wrote that leukocytes, neutrophils, and CRP are usually high. Is that always the case? Please write in this part of the manuscript, according to the available literature, the sensitivity and specificity of the above tests when we talk about appendicitis in children.

It would be good to know the exact time frame before surgery when the blood was sampled to measure IL-6.

Please write exactly from whom the informed consents were obtained; from parents/legal guardians or patients themselves?

As part of the table, it would be good if you could write the meaning of all abbreviations.

Instead of DS you probably meant SD.

Please put a sign (*) next to all p-values that are statistically significant in the table.

It is very important to write down all the conditions in which IL-6 can be elevated in children. Did you think about this fact when writing the manuscript? If not, please add the same to the study limitation.
